# LncKCNQ1OT1 Promotes the Odontoblastic Differentiation of Dental Pulp Stem Cells via Regulating hsa-miR-153-3p/RUNX2 Axis

**DOI:** 10.3390/cells11213506

**Published:** 2022-11-05

**Authors:** Xiaohui Lu, Jiawen Zhang, Yuanzhou Lu, Jing Xing, Min Lian, Guijuan Feng, Dan Huang, Chenfei Wang, Nimei Shen, Xingmei Feng

**Affiliations:** 1Department of Stomatology, Affiliated Hospital of Nantong University, Nantong 226001, China; 2Department of Cardiology, Tongzhou People’s Hospital, Nantong 226399, China; 3Department of Ophthalmology and Otorhinolaryngology, The Second Affiliated Hospital of Nantong University, Nantong 226001, China

**Keywords:** LncKCNQ1OT1, odontoblastic differentiation, dental pulp stem cells, hsa-miR-153-3p, RUNX2

## Abstract

This study aimed to explore the role of LncKCNQ1OT1/hsa-miR-153-3p/RUNX2 in the odontoblastic differentiation of human dental pulp stem cells (DPSCs) and its possible mechanism. The expression of LncKCNQ1OT1, hsa-miR-153-3p, and RUNX2 in the odontoblastic differentiation was detected by qRT-PCR. Interaction between LncKCNQ1OT1 and hsa-miR-153-3p and interaction between hsa-miR-153-3p and RUNX2 were detected by dual-luciferase assay. The cell viability of DPSCs was detected by CCK-8, and the effect of LncKCNQ1OT1 and hsa-miR-153-3p on the odontoblastic differentiation of DPSCs was observed by alizarin red staining, alkaline phosphatase (ALP) activity assay, and Western blot for RUNX2, DSPP, and DMP-1. The results showed, during odontoblastic differentiation of DPSCs, the expression of LncKCNQ1OT1 increased, hsa-miR-153-3p expression decreased, and RUNX2 expression increased. Dual-luciferase assay showed that LncKCNQ1OT1 sponges hsa-miR-153-3p and hsa-miR-153-3p targets on RUNX2. After LncKCNQ1OT1 and hsa-miR-153-3p expressions of DPSCs were changed, the cell viability was not notably changed, but the odontoblastic differentiation was notably changed, which was confirmed with Alizarin Red staining, ALP activity, and Western blot for RUNX2, DSPP, and DMP-1. The results indicate that LncKCNQ1OT1 promotes the odontoblastic differentiation of DPSCs via regulating hsa-miR-153-3p/RUNX2 axis, which may provide a therapeutic clue for odontogenesis.

## 1. Introduction

Human dental pulp stem cells (DPSCs) are a kind of adult stem cell with high proliferation, self-renewal ability, and multi-differentiation potential, which are mainly derived from dental pulp tissue. DPSCs can not only differentiate into dentin, but also into bone, cartilage, fat, myogenic, and other mesodermal-derived cells [1,2]. Human DPSCs are usually taken from extracted third molars or healthy premolars that need to be extracted for orthodontics. The DPSCs are easier to obtain. Therefore, DPSCs are ideal seed cells for tissue engineering [3,4,5]. The ability of DPSCs to differentiate into odontoblasts plays an important role in maintaining the dynamic balance of dental pulp tissue and tooth regeneration, and is the basis for future use of tissue engineering to achieve dentin regeneration [6,7,8]. The regulation of odontoblastic differentiation of DPSCs is still difficult and is a research hotspot. Odontoblastic differentiation of DPSCs involves many factors, such as biological scaffolds, regulatory genes, and signal pathways [8,9,10,11,12]. MicroRNAs (miRNAs, miRs), as post-transcriptional inhibitors, recognize and bind to the 3′ untranslated region (3′UTR) of the target gene to inhibit the translation of the target gene protein and have complex regulatory effects on the body’s physiological/pathological activities, including the process of odontoblastic differentiation [13,14,15]. Long noncoding RNAs (LncRNAs), as competing endogenous RNAs (ceRNAs), play a key role in cell cycle, migration, proliferation, differentiation, and apoptosis through sponging miRNAs to regulate miRNA targets [16,17]. More and more studies have shown that LncRNAs participate in the differentiation process of odontoblasts through sponging miRNAs [10,18,19,20,21,22].

Studies have shown that runt-related transcription factor 2 (RUNX2) plays a key role in the odontoblastic differentiation. Its expression promotes the odontoblastic differentiation [23,24,25]. In our study, the software targetscan (http://www.targetscan.org, accessed on 1 September 2021) was used to analyze which miRNAs may target the RUNX2, and the result showed RUNX2 may be targeted by hsa-miR-153-3p. We used LncBase Predicted v.2 database (http://carolina.imis.athena-innovation.gr, accessed on 1 September 2021) to analyze the potential LncRNAs which interact with hsa-miR-153-3p, and the results showed LncKCNQ1OT1 contains a potential binding site of hsa-miR-153-3p. We chose these two genes as our research targets because they have been reported to be involved in the osteogenic differentiation process, and this process shares some common key regulatory genes, such as RUNX2, with odontoblastic differentiation [24,26,27,28]. However, the odontoblastic differentiation is different from osteogenic differentiation because the dentin and bone are different tissues. The interaction between LncKCNQ1OT1 and hsa-miR-153-3p or hsa-miR-153-3p and RUNX2, and their roles in the differentiation of DPSCs into odontoblasts are still unclear. We speculate LncKCNQ1OT1 promotes the odontoblastic differentiation of DPSCs via regulating hsa-miR-153-3p/RUNX2 axis. Therefore, the study aimed to explore the effect of LncKCNQ1OT1/hsa-miR-153-3p/RUNX2 on the odontoblastic differentiation of DPSCs and confirm its regulation axis.

## 2. Materials and Methods

### 2.1. DPSCs Culture 

This study was approved by the Ethics Committee of the Affiliated Hospital of Nantong University (2015-018, 9 March 2015). We declared that all methods were carried out in accordance with the relevant guidelines and regulations. The cell culture and identification were performed as described in our previous studies [13]. Briefly, the pulp from normal human impacted third molars was collected and digested with 3 mg/mL collagenase type I for 1 h at 37 °C. Eight donors of the impacted third molar (10 teeth) had given informed consent. Single-cell suspensions of dental pulp were cultured and passaged in Dulbecco modified Eagle medium (DMEM) supplemented with 10% fetal bovine serum (FBS), 100 U/mL penicillin, and 100 μg/mL streptomycin at 37 °C under 5% CO_2_. The fourth passage cells were used in the following experiments. 

### 2.2. Cell Transfection

DPSCs were cultured in a 6-well culture plate at a concentration of 5 × 10^3^ /mL. After 24 h, they were transfected with KCNQ1OT1 siRNA, siRNA-NC (RuiboBio; Changzhou, China), overexpression vectors of KCNQ1OT1 (pcDNA-KCNQ1OT1) (Sangon Biotech; Shanghai, China), hsa-miR-153-3p inhibitor, inhibitor–NC, hsa-miR-153-3p mimic, and mimic NC (ThemoFisher; Waltham, MA, USA) with lipofectamine 2000 (ThemoFisher; Waltham, MA, USA) according to the transfection instructions; the cells were incubated in a 5% CO_2_, 37 °C incubator and a blank control group was set up. After 48 h, the transfected cells were taken for the following experiment.

### 2.3. Cell Viability Assay

The cell counting Kit-8 (CCK-8; Beyotime Biotechnology, Haimen, China) was used to detect the viability of the DPSCs according to the manufacturer instructions, and the OD value was read at 450 nm.

### 2.4. Odontoblastic Differentiation Culture

The odontoblastic differentiation culture was performed as described in our previous study [13]. Briefly, the fourth passage DPSCs were cultured in odontogenic differentiation medium containing a minimum essential medium (Invitrogen, Carlsbad, CA, USA), 50 mg/mL a-ascorbic acid, 15% FBS, 10 mmol/L b-glycerophosphate, 10 nmol/L dexamethasone (Sigma-Aldrich, St Louis, MO, USA), 0.292 mg/mL glutamine, 100 μg/mL streptomycin, and 100 U/mL penicillin for 14 days.

### 2.5. Quantitative Real-Time PCR (qRT-PCR)

The total RNA in cells was extracted using Trizol (Invitrogen, Carlsbad, CA, USA) and quantified with agarose gel (Sigma-Aldrich, St Louis, MO, USA). For LncRNA and mRNA, cDNA was synthesized using TransScript First-Strand cDNA Synthesis SuperMix (Transgen, Beijing, China). For miRNA, cDNA was synthesized using TransScript miRNA First-Strand cDNA Synthesis SuperMix (Transgen, Beijing, China). GAPDH and U6 were used as the internal reference. The gene expression level was calculated with 2^−ΔΔCt^ method. The primer sequences are shown in the Table 1.

### 2.6. Luciferase Reporter Gene Experiment

The wild-type (WT) binding fragment of LncKCNQ1OT1 or WT core sequence of 3′UTR of RUNX2 was cloned into pMIR-Report luciferase vector (Promega Corporation, Madison, WI, USA) to construct WT-LncKCNQ1OT1 reporter vector or WT-RUNX2 reporter vector. The fragment of LncKCNQ1OT1 or core sequence of 3′UTR of RUNX2 were mutated (MU) and cloned into pMIR-Report luciferase vector to obtain MU-LncKCNQ1OT1 reporter vector or MU-RUNX2 reporter vector. The WT-LncKCNQ1OT1, MU-LncKCNQ1OT1, and hsa-miR-153-3p mimic (ThemoFisher, Waltham, MA, USA) or NC were co-transfected into HEK293 cells to investigate the relationship between LncKCNQ1OT1 and hsa-miR-153-3p. The WT-RUNX2, MU-RUNX2, and hsa-miR-153-3p mimic (ThemoFisher, Waltham, MA, USA) or NC were co-transfected into HEK293 cells to investigate the relationship between hsa-miR-153-3p and RUNX2. After 48 h following transfection, relative luciferase activity was analyzed with a dual-luciferase reporter assay system (Promega Corporation, Madison, WI, USA) according to the manufacturer instructions. 

### 2.7. Alkaline Phosphatase (ALP) Activity Assay

After 14 days of odontoblastic differentiation, the cells were lysed with 1% Triton X-100 for 15 min and centrifuged at 10,000× *g* for 5 min; then, the supernatant was collected and detected with ALP Assay Kit (Beyotime Biotechnology, Haimen, China) according to the manufacturer instructions, and the OD value was read at 405 nm. 

### 2.8. Alizarin Red Staining

After 14 days of odontoblastic differentiation, the cells were incubated with 2% alizarin red staining solution (Beyotime Biotechnology, Haimen, China) for 10 min at room temperature. Then, the cells were observed under an inverted microscope and the cell mineralization was quantified with alizarin red extracted with 100 mM cetylpyridinium chloride solution (Sigma, St Louis, MO, USA), and the OD value was read at 570 nm.

### 2.9. Western Blot Analysis

After 14 days of odontoblastic differentiation, the total protein of cells was extracted by RIPA buffer (Beyotime Biotechnology, Haimen, China). The protein was electrophoresed and transferred to PVDF membrane. After PVDF membrane was blocked with 5% BSA, it was successively incubated with the primary antibody: rabbit anti-RUNX2 primary antibody (1:800, Abcam, Cambridge, UK), rabbit anti-DSPP (1:1000, Abcam, Cambridge, UK), rabbit anti-DMP-1 (1:1000, ABclonal, Wuhan, China), or mouse anti-β-actin (1:1000, Abcam, Cambridge, UK) and the second antibody: IRDye 700-conjugated affinity-purified goat anti-mouse (1:4000, Rockland Immunochemicals, Philadelphia, Pennsylvania, PA, USA) or IRDye 800-conjugated affinity-purified goat anti-rabbit second antibody (1:4000, Rockland Immunochemicals, Philadelphia, Pennsylvania, PA, USA). The relative protein expression level was analyzed with Odyssey laser scanning system (LI-COR Inc., Lincoln, NE, USA).

### 2.10. Statistical Analysis

The SPSS 22.0 software (IBM, Chicago, IL, USA) was used to analyze the data of this research. The data of this study are presented as mean ± standard deviation (M ± SD). Comparison among groups was tested with independent t test or one-way analysis of variance (ANOVA) followed by Tukey’s test. The Pearson analysis was used to analyze the correlation. A statistical graph was made using GraphPad Prism 7 software. *p* < 0.05 was considered statistically significant.

## 3. Results

### 3.1. LncKCNQ1OT1, hsa-miR-153-3p, and RUNX2 Expression Levels during Odontoblastic Differentiation

During the odontoblastic differentiation of DPSCs, the expressions of LncKCNQ1OT1, hsa-miR-153-3p, and RUNX2 were detected at 0, 1, 3, 7, and 14 d. The relative expression levels of LncKCNQ1OT1, hsa-miR-153-3p, and RUNX2 were quantified with the expression level at 0 d when the DPSCs did not begin the odontoblastic differentiation. On 1 d of differentiation of odontoblasts, the expressions of LncKCNQ1OT1 and RUNX2 increased, and hsa-miR-153-3p expression level decreased, and they reached a peak at 3 d; then, their expression levels at 7 and 14 d were similar to those at 3 d (*F* = 20.54, *p* < 0.01; *F* = 53.61, *p* < 0.01; *F* = 174.87) (Figure 1a–c). Spearman correlation analysis showed that LncKCNQ1OT1 and RUNX2 expression levels were positively correlated (*r* = 0.72, *p* < 0.01); LncKCNQ1OT1 and hsa-miR-153-3p expression levels were negatively correlated (*r* = −0.80, *p* < 0.01); hsa-miR-153-3p and RUNX2 expression levels were negatively correlated (*r* = −0.89, *p* < 0.01) (Figure 1d–f).

### 3.2. The Effect of LncKCNQ1OT1 on the Cell Viability and Odontoblastic Differentiation of DPSCs

After pcDNA-KCNQ1OT1 or KCNQ1OT1 siRNA transfection, the LncKCNQ1OT1 expression level was detected by qRT-PCR. LncKCNQ1OT1 expression level of DPSCs transfected with pcDNA-KCNQ1OT1 significantly increased and that of DPSCs transfected with KCNQ1OT1 siRNA notably decreased (*F* = 141.41, *p* < 0.01) (Figure 2a). The viability of DPSCs was assessed with CCK-8 kit. Compared with the control, the differences in OD value among groups were not statistically significant (*F* = 1.77, *p* = 0.16). The transfection did not affect the viability of DPSCs (Figure 2b).

After 14 days of culture, the odontoblastic differentiation of DPSCs was detected by ALP activity, alizarin red staining and Western blot for RUNX2, DSPP, and DMP-1. The result of ALP activity assay showed the OD value of the pcDNA-KCNQ1OT1 group was higher than that of control and NC groups, while the OD value of the KCNQ1OT1 siRNA group was lower than that of control and NC groups (*F* = 87.56, *p* < 0.01) (Figure 2c). The result of alizarin red staining displayed that mineralized bone matrix increased in the pcDNA-KCNQ1OT1 group and decreased in the KCNQ1OT1 siRNA group compared with the control and NC groups (Figure 2d). The result of Western blot showed the protein expression levels of RUNX2, DSPP, and DMP-1 in the pcDNA-KCNQ1OT1 group increased and in the KCNQ1OT1 siRNA group decreased compared with the control and NC groups (*F* = 100.49, *p* < 0.01; *F* = 76.02, *p* < 0.01; *F* = 63.67, *p* < 0.01) (Figure 2e).

### 3.3. Overexpression of hsa-miR-153-3p Reverses the Promotion of Odontoblastic Differentiation Induced by LncKCNQ1OT1

After pcDNA-KCNQ1OT1, pcDNA-KCNQ1OT1+mimic or pcDNA-KCNQ1OT1+mimic NC transfection, LncKCNQ1OT1 and hsa-miR-153-3p expression levels were detected by qRT-PCR. LncKCNQ1OT1 expression level of DPSCs transfected with pcDNA-KCNQ1OT1, pcDNA-KCNQ1OT1+mimic or pcDNA-KCNQ1OT1+mimic NC significantly increased (*F* = 25.57, *p* < 0.01) (Figure 3a), and hsa-miR-153-3p expression level of DPSCs transfected pcDNA-KCNQ1OT1 or pcDNA-KCNQ1OT1+mimic NC significantly decreased, and hsa-miR-153-3p expression level of DPSCs transfected with pcDNA-KCNQ1OT1+mimic was similar to the control group (*F* = 16.83, *p* < 0.01) (Figure 3b). In addition, the results of CCK-8 assay showed the transfection did not affect the viability of DPSCs (*F* = 0.69, *p* = 0.57) (Figure 3c).

After 14 days of odontoblastic differentiation, the mineralized bone matrix, ALP activity and protein expression levels of RUNX2, DSPP, and DMP-1 in the pcDNA-KCNQ1OT1 group increased compared with the control group. While DPSCs were co-transfected with pcDNA-KCNQ1OT1 and hsa-miR-153-3p mimic, the mineralized bone matrix, ALP activity, RUNX2, DSPP, and DMP-1 protein expression levels were close to the control group. When DPSCs were co-transfected with pcDNA-KCNQ1OT1 and mimic NC, the mineralized bone matrix, ALP activity, RUNX2, DSPP, and DMP-1 protein expression levels were similar to the pcDNA-KCNQ1OT1 group (*F* = 82.96, *p* < 0.01; *F* = 12.51, *p* < 0.01; *F* = 13.45, *p* < 0.01) (Figure 3d–f).

### 3.4. LncKCNQ1OT1 Act as a Sponge of hsa-miR-153-3p and hsa-miR-153-3p Target on RUNX2

Luciferase reporter gene experiment was used to verify that LncKCNQ1OT1 acts as a sponge of hsa-miR-153-3p and hsa-miR-153-3p target on RUNX2. The result showed that, when the binding fragment of LncKCNQ1OT1 or core sequence of 3′UTR of RUNX2 was mutated, the luciferase relative activity of mimic group was similar to the NC group (*t* = 0.86, *p* = 0.41; *t* = 1.09, *p* = 0.30). However, in the WT-LncKCNQ1OT1 or WT-RUNX2 reporter gene system, the relative activity of luciferase in mimic group was significantly lower than that in NC group (*t* = 7.20, *p* < 0.01; *t* = 7.20, *p* < 0.01) (Figure 4a,b).

## 4. Discussion

Odontoblasts are terminally differentiated cells from DPSCs and are one of the main cells that form dental tissues. Odontoblastic differentiation is the prerequisite for dentin formation. Studying the influence of various signaling pathways on the differentiation of odontoblasts, regulating the expression of various signaling pathways, and promoting the differentiation of odontoblasts will be of great significance for the treatment of various dentin-related diseases. More and more research proves that LncRNAs and miRNAs play key roles in odontoblastic differentiation [10,18,19]. In our study, we found that LncKCNQ1OT1 promoted the odontoblastic differentiation of DPSCs via regulating hsa-miR-153-3p/RUNX2 axis.

RUNX2 is one of the members of the Runt family, which is a specific transcription factor of odontoblastic differentiation of DPSCs [23]. Studies showed the expression of RUNX2 was involved in the differentiation of odontoblasts and osteoblasts lining bone in the periodontal space [29,30]. RUNX2- mice exhibited molar developmental arrest at the early cap stage, suggesting that RUNX2 was required in the progression of tooth development from the cap stage to the bell stage [29]. Therefore, the regulation of RUNX2 was important in the odontoblastic differentiation. In our study, during the odontoblastic differentiation of DPSCs, the expression of RUNX2 increased, which is consistent with the research results of other scholars [31]. Studies showed many miRNAs regulated the odontoblastic differentiation via targeting on RUNX2 [15,32,33]. In our study, we used targetscan software to find whether hsa-miR-153-3p has the binding site of RUNX2 3′UTR. Jiang et al. reported that hsa-miR-153-3p inhibited osteogenic differentiation of periodontal ligament stem cells via targeting KDM6A and regulating the ALP, RUNX2, and OPN transcription [27]. The odontoblastic differentiation process is similar to the osteogenic differentiation [24,26,27,28,29]. Therefore, in our study, the expression level of hsa-miR-153-3p during odontoblastic differentiation of DPSCs was detected and the result showed it decreased in the process and hsa-miR-153-3p expression level was negatively correlated with RUNX2 expression level. The luciferase reporter gene experiment confirmed that RUNX2 is a target of hsa-miR-153-3p. The results indicated that hsa-miR-153-3p was indeed involved in the odontoblastic differentiation of DPSCs, and it was a negative regulatory factor.

The mechanism by which the expression of hsa-miR-153-3p decreases during the differentiation of odontoblasts is still unclear. LncRNAs as ceRNAs can sponge miRNAs. We used LncBase Predicted v.2 database to find whether LncKCNQ1OT1 contained the potential binding site of hsa-miR-153-3p. Study showed that knockdown of LncKCNQ1OT1 in tendon stem cell inhibited the osteogenic differentiation [34]. The result of qRT-PCR in our study showed LncKCNQ1OT1 increased during odontoblastic differentiation of DPSCs and its expression level was negatively correlated with hsa-miR-153-3p. The luciferase reporter gene experiment confirmed that LncKCNQ1OT1 sponges hsa-miR-153-3p. The results indicated that LncKCNQ1OT1 positively regulated the odontoblastic differentiation of DPSCs. Therefore, we constructed pcDNA-KCNQ1OT1 or KCNQ1OT1 siRNA to transfect DPSCs to enhance or downregulate LncKCNQ1OT1 expression level. The result of CCK-8 showed LncKCNQ1OT1 expression changes had no effect on the cell viability. The odontoblastic differentiation of DPSCs was detected by alizarin red staining, ALP activity, and Western blot for RUNX2, DSPP, and DMP-1, and the findings showed downregulated LncKCNQ1OT1 expression inhibited the odontoblastic differentiation of DPSCs, while LncKCNQ1OT1 overexpression promoted the odontoblastic differentiation of DPSCs. When DPSCs were co-transfected with pcDNA-KCNQ1OT1 and hsa-miR-153-3p mimic, DPSCs overexpressed LncKCNQ1OT1 and hsa-miR-153-3p at the same time; LncKCNQ1OT1’s promotion of odontoblast differentiation was reversed.

## 5. Conclusions

In summary, our data demonstrated that LncKCNQ1OT1 promotes the odontoblastic differentiation of DPSCs via regulating hsa-miR-153-3p/RUNX2 axis, which may provide a therapeutic clue for odontogenesis in the dental tissue engineering.

## Figures and Tables

**Figure 1 cells-11-03506-f001:**
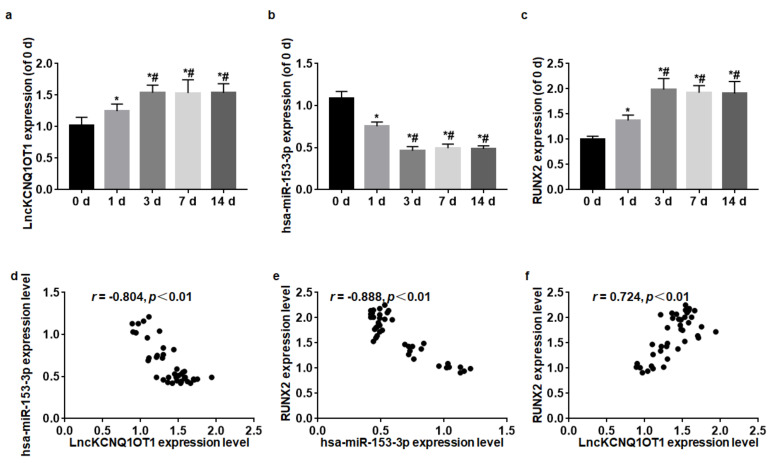
LncKCNQ1OT1, hsa-miR-153-3p, and RUNX2 expression levels during odontoblastic differentiation were detected by qRT-PCR. (**a**) On 1 d of differentiation of odontoblasts, the expression of LncKCNQ1OT1 increased and reached a peak at 3 d. (**b**) On 1 d of differentiation of odontoblasts, hsa-miR-153-3p expression level decreased and reached a peak at 3 d. (**c**) On 1 d of differentiation of odontoblasts, the expression of RUNX2 increased and reached a peak at 3 d. (**d**) LncKCNQ1OT1 expression level was negatively correlated with hsa-miR-153-3p expression level. (**e**) hsa-miR-153-3p expression level was negatively correlated with RUNX2 expression level. (**f**) LncKCNQ1OT1 expression level was positively correlated with RUNX2 expression level. * vs. 0 d, *p* < 0.05; # vs. 1 d, *p* < 0.05.

**Figure 2 cells-11-03506-f002:**
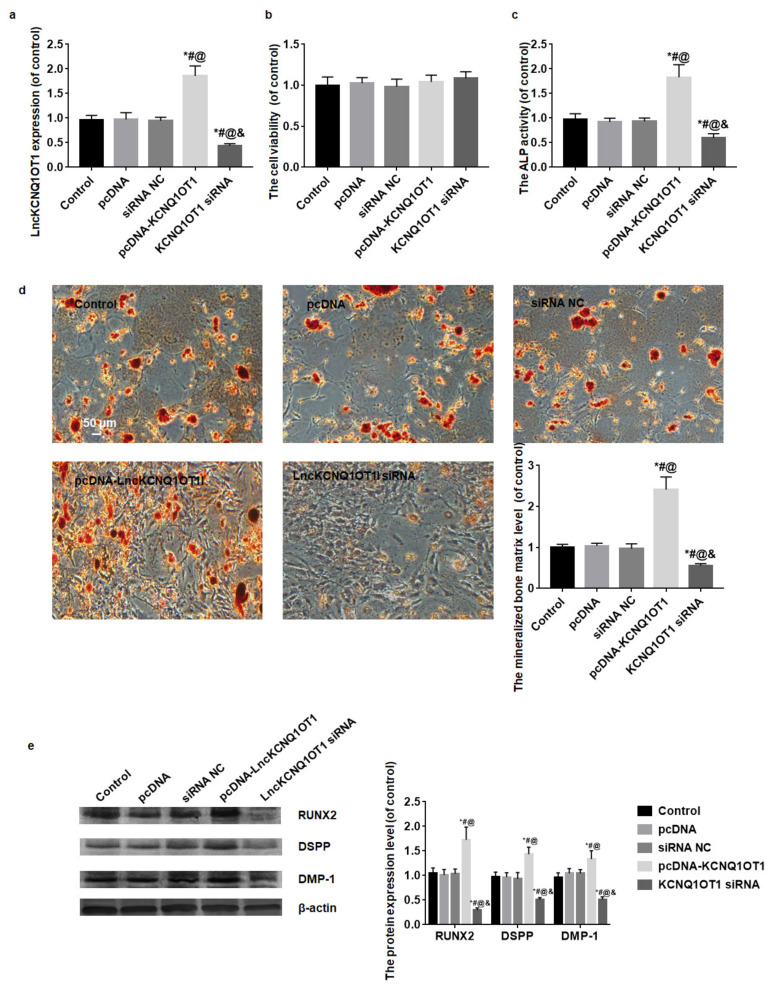
The effect of LncKCNQ1OT1 on the cell viability and odontoblastic differentiation of DPSCs. (**a**) LncKCNQ1OT1 expression level of DPSCs transfected with pcDNA-KCNQ1OT1 significantly increased and that of DPSCs transfected with KCNQ1OT1 siRNA notably decreased. (**b**) The transfection did not affect the viability of DPSCs. (**c**) After 14 days of odontoblastic differentiation, ALP activity in the pcDNA-KCNQ1OT1 group increased and in the KCNQ1OT1 siRNA group decreased compared with the control and NC groups. (**d**) After 14 days of odontoblastic differentiation, the mineralized bone matrix in the pcDNA-KCNQ1OT1 group increased and in the KCNQ1OT1 siRNA group decreased compared with the control and NC groups. (**e**) After 14 days of odontoblastic differentiation, RUNX2, DSPP, and DMP-1 protein expression levels in the pcDNA-KCNQ1OT1 group increased and in the KCNQ1OT1 siRNA group decreased compared with the control and NC groups. Bar = 50 μm. * vs. control group, *p* < 0.05; # vs. pcDNA group, *p* < 0.05; @ vs. siRNA NC group, *p* < 0.05; & vs. pcDNA-KCNQ1OT1 group, *p* < 0.05.

**Figure 3 cells-11-03506-f003:**
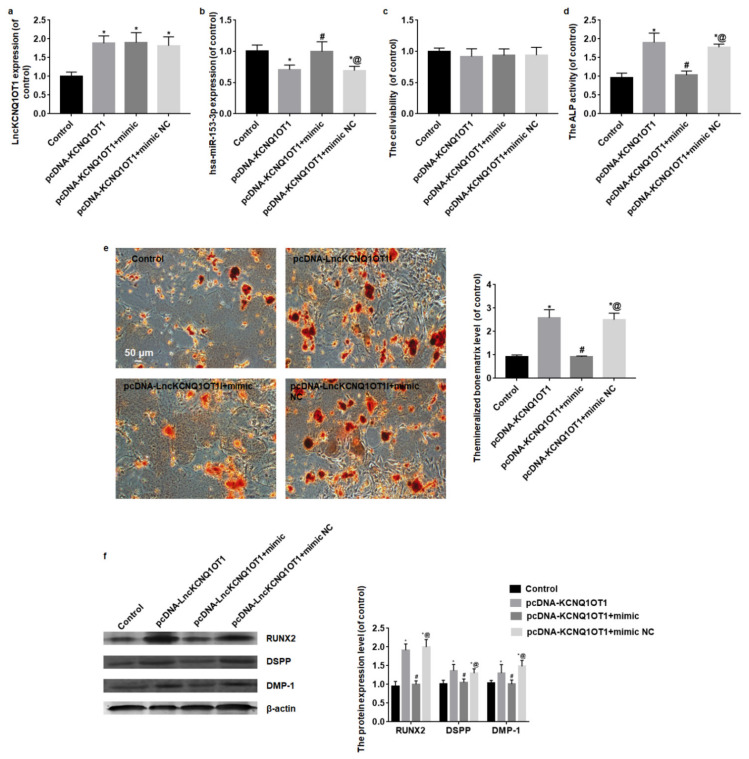
Overexpression of hsa-miR-153-3p reverses the promotion of odontoblastic differentiation induced by LncKCNQ1OT1. (**a**) LncKCNQ1OT1 expression level of DPSCs transfected with pcDNA-KCNQ1OT1, pcDNA-KCNQ1OT1+mimic or pcDNA-KCNQ1OT1+mimic NC significantly increased. (**b**) hsa-miR-153-3p expression level of DPSCs transfected pcDNA-KCNQ1OT1 or pcDNA-KCNQ1OT1+mimic NC significantly decreased, and hsa-miR-153-3p expression level of DPSCs transfected with pcDNA-KCNQ1OT1+mimic was similar to the control group. (**c**) The results of CCK-8 assay showed the transfection did not affect the viability of DPSCs. (**d**–**f**) After 14 days of odontoblastic differentiation, the ALP activity, mineralized bone matrix, RUNX2, DSPP, and DMP-1 protein expression levels in the pcDNA-KCNQ1OT1 group increased comparison with the control group. While DPSCs were co-transfected with pcDNA-KCNQ1OT1 and hsa-miR-153-3p mimic the ALP activity, mineralized bone matrix, RUNX2, DSPP, and DMP-1 protein expression levels were close to the control group. Bar = 50 μm. * vs. control group, *p* < 0.05; # vs. pcDNA-KCNQ1OT1 group, *p* < 0.05; @ vs. pcDNA-KCNQ1OT1+mimic group, *p* < 0.05.

**Figure 4 cells-11-03506-f004:**
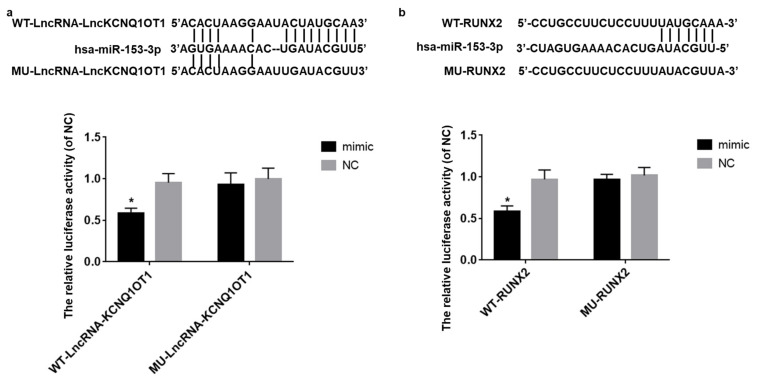
LncKCNQ1OT1 act as a sponge of hsa-miR-153-3p and hsa-miR-153-3p target on RUNX2. (**a**) In the WT-LncKCNQ1OT1 reporter gene system, the relative activity of luciferase in mimic group was significantly lower than that in NC group. When the binding fragment of LncKCNQ1OT1 was mutated, the luciferase relative activity of mimic group was similar to the NC group. (**b**) In the WT-RUNX2 reporter gene system, the relative activity of luciferase in mimic group was significantly lower than that in NC group. When the core sequence of 3′UTR of RUNX2 was mutated, the luciferase relative activity of mimic group was similar to the NC group. * vs. NC group, *p* < 0.05.

**Table 1 cells-11-03506-t001:** The primer sequences for qRT-PCR.

Gene	Primer Sequences (5′-3′)
LncKCNQ1OT1	Forward: ACTCACTCACTCACTCACT
	Reverse: CTGGCTCCTTCTATCACATT
hsa-miR-153-3p	Forward: ACACTCCAGCTGGGTTGCATAGTCACAAA
	Reverse: CAGTGCGTGTCGTGGAGT
RUNX2	Forward: TGCCACCTCTGACTTCTGC
	Reverse: GATGAAATGCCTGGGAACTG
U6	Forward: GTGCTCGCTTCGGCAGCACAT
	Reverse: TACCTTGCGAAGTGCTTAAAC
GAPDH	Forward: AGGTGAAGGTCGGAGTCAAC
	Reverse: CGCTCCTGGAAGATGGTGAT

## Data Availability

The datasets used and/or analyzed during the current study are available from the corresponding author on reasonable request.

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
