# Peer review of "LncKCNQ1OT1 Promotes the Odontoblastic Differentiation of Dental Pulp Stem Cells via Regulating hsa-miR-153-3p/RUNX2 Axis"

_cells, 2022, doi:10.3390/cells11213506_

Round 1

Reviewer 1 Report

In the manuscript, the authors aimed to explore the role of LncRNA-KCNQ1OT in the odontoblastic differentiation of human dental pulp stem cells (hDPSCs), which are a special type of mesenchymal stem cell. For this, following identify that the LncRNA target the miR-183-3p (a microRNA that target the RUNX2 3'UTR, which is a key mediator of odontoblastic differentiation), they transfected DPSCs with LncRNA. Results obtained provided strong evidence that the LncRNA-KCNQ1OT has a key play in osteogenic differentiation of DPSCs. This result is very important, since there are few iformation about the role of non-coding RNA in differentiation process of DPSCs

Author Response

Thank you so much for your comments.

Reviewer 2 Report

The authors investigated the role of lncKCNQ1OT1 in DPSCs odontoogenic differentiation, however, the research has major concerns. 

1. All figures are too small to read. The reviewer suggests enlarge font size in all figures.  

2. In Figs. 2 and 3, Alizarin red staining does not seem work as 14 days of osteogenic differentiation should lead to more calcium deposit. 

3. Y axis of gene expression should be shortened, for example, LncKCNQ1OT1 mRNA expression. 

4.  miR-153-3p should be full name, has-miR-153-3p. LncKCNQ1OT1 can be appropriate. 

5.  DMEM and osteogenic differentiation of all genes should be included. 

6. Figure 4: the osteogenic and odontogenic differentiation markers should be included. 

Author Response

  1. All figures are too small to read. The reviewer suggests enlarge font size in all figures.

Answer: It has been enlarged. Thank you so much!

  1. In Figs. 2 and 3, Alizarin red staining does not seem work as 14 days of osteogenic differentiation should lead to more calcium deposit.

Answer: This result is true. It may be related to cell density and cell state. Thank you so much!

  1. Y axis of gene expression should be shortened, for example, LncKCNQ1OT1 mRNA expression.

Answer: It has been revised. Thank you so much!

  1. miR-153-3p should be full name, has-miR-153-3p. LncKCNQ1OT1 can be appropriate.

Answer: It has been revised. Thank you so much!

  1. DMEM and osteogenic differentiation of all genes should be included.

Answer: Our research mainly focuses on the odontoblastic differentiation of dental pulp stem cells. The role of this axis in the osteoblast lineage cells is still unclear which will be confirmed in the future study. Thank you so much!

  1. Figure 4: the osteogenic and odontogenic differentiation markers should be included.

Answer: DSPP, DMP-1and RUNX2 have been detected in the Figure-2,3. Figure 4 showed Luciferase reporter gene experiment confirmed this axis. Thank you so much!

Thank you for the advices on my paper. I have profited so much from your proposal.

Reviewer 3 Report

To the authors:

The authors focused on the regulatory mechanisms of odontogenic differentiation via LncRNA and miRNA, approached by in silico data analyses. Then the authors found LncRNA-KCNQ1OT1 negatively regulated miR-153-3p as RNA sponge, to avoid the suppression of Runx2 expression which is one of the targets of miR-153-3p. Then they concluded this axis promotes odontogenic differentiation of DPSCs. The results are clear, but the several information is missing or unclear description. Is this axis already known in osteoblasts? What is new and your perspective in the future based on this finding?  Therefore, the reviewer recommends revised manuscript.

1.     On line 63-65 in page 2, the authors states “We chose these two genes as our research targets because they have been reported to be involved in the osteogenic differentiation process and this process share some common key regulatory genes, such as RUNX2 with odontoblastic differentiation.”. However, they did not referred any publications. Please add the references.

2.     Through the results section, it seems the first sentence is difficult to understand. Please write the object. For example, on Line189 in page 5, “After transfection, the LncRNA-KCNQ1OT1 expression level was detected by qRT-PCR.” After transfection of what? Please describe the reader friendly to avoid misunderstanding. This kind of sentences are often found.

3.     Is this LncRNA-KCNQ1OT1, miR-153-3p, Runx2 axis unique in the odontoblast lineage cells? 

Or, it is shared with osteoblast lineage cells? 

4.     Related with the previous question. On Line 298-299 in page 9, the authors states ”The odontoblastic differentiation process is similar to the osteogenic differentiation.” They need references.

5.     Overall data is clear, but it is not clear the difference between osteoblast differentiation and odontoblast differentiation. Is this the common mechanism for the mineralization steps?

6.     The authors mentioned “may provide a therapeutic clue for odontogenesis.” on 327-328 in page 9. For example, what kind of therapeutic things can be imagined?  

Author Response

  1. On line 63-65 in page 2, the authors states “We chose these two genes as our research targets because they have been reported to be involved in the osteogenic differentiation process and this process share some common key regulatory genes, such as RUNX2 with odontoblastic differentiation.”. However, they did not referred any publications. Please add the references.

Answer: It has been added. Thank you so much!

  1. Through the results section, it seems the first sentence is difficult to understand. Please write the object. For example, on Line189 in page 5, “After transfection, the LncRNA-KCNQ1OT1 expression level was detected by qRT-PCR.” After transfection of what? Please describe the reader friendly to avoid misunderstanding. This kind of sentences are often found.

Answer: It has been revised. Thank you so much!

  1. Is this LncRNA-KCNQ1OT1, miR-153-3p, Runx2 axis unique in the odontoblast lineage cells? Or, it is shared with osteoblast lineage cells?

Answer: Our results indicated this axis play a key role in the odontoblastic differentiation. Its role in the osteoblast lineage cells is still unclear which will be confirmed in the future study. Thank you for providing us with valuable research ideas.

  1. Related with the previous question. On Line 298-299 in page 9, the authors states ”The odontoblastic differentiation process is similar to the osteogenic differentiation.” They need references.

Answer: It has been added. Thank you so much!

  1. Overall data is clear, but it is not clear the difference between osteoblast differentiation and odontoblast differentiation. Is this the common mechanism for the mineralization steps?

Answer: Odontoblastic differentiation is similar to osteogenic differentiation because they share some common key regulatory genes, such as RUNX2, and maybe share the common mechanism for the mineralization steps. The differences between osteogenic and odontoblastic differentiation are still unclear which need be studied in the following days. Thank you so much!

  1. The authors mentioned “may provide a therapeutic clue for odontogenesis.” on 327-328 in page 9. For example, what kind of therapeutic things can be imagined?

Answer: It has been added. Thank you so much!

Thank you for the advices on my paper. I have profited so much from your proposal.

Round 2

Reviewer 2 Report

The authors addressed most of the comments. This work is suitable for Special Issue ' Epigenetics in Periodontal Disease diagnosis, systemic interactions and treatment'.